# Mechanism of Action of Peripheral Nerve Stimulation for Chronic Pain: A Narrative Review

**DOI:** 10.3390/ijms24054540

**Published:** 2023-02-25

**Authors:** Lady Christine Ong Sio, Brian Hom, Shuchita Garg, Alaa Abd-Elsayed

**Affiliations:** 1Department of Anesthesiology, University of Cincinnati College of Medicine, Cincinnati, OH 45267, USA; 2Department of Anesthesiology, University of Wisconsin School of Medicine and Public Health, Madison, WI 53726, USA

**Keywords:** peripheral nerve stimulation, neuromodulation, chronic pain, mechanism of action

## Abstract

The use of stimulation of peripheral nerves to test or treat various medical disorders has been prevalent for a long time. Over the last few years, there has been growing evidence for the use of peripheral nerve stimulation (PNS) for treating a myriad of chronic pain conditions such as limb mononeuropathies, nerve entrapments, peripheral nerve injuries, phantom limb pain, complex regional pain syndrome, back pain, and even fibromyalgia. The ease of placement of a minimally invasive electrode via percutaneous approach in the close vicinity of the nerve and the ability to target various nerves have led to its widespread use and compliance. While most of the mechanism behind its role in neuromodulation is largely unknown, the gate control theory proposed by Melzack and Wall in the 1960s has been the mainstay for understanding its mechanism of action. In this review article, the authors performed a literature review to discuss the mechanism of action of PNS and discuss its safety and usefulness in treating chronic pain. The authors also discuss current PNS devices available in the market today.

## 1. Introduction

It is estimated that about 10% of patients who suffer from chronic pain may have a component of peripheral nerve injury [1]. A subsequent change in the chemical milieu can cause persistent and aberrant nociception and this triggers the cascade of chemical and conformational changes at both the spinal and supraspinal levels, leading to chronic neuropathic pain [2].

While the use of conservative techniques and tools to manage pain symptoms still play a significant role in chronic pain management, there has been a keen interest recently in applying the concept of neuromodulation, called peripheral nerve stimulation (PNS), in this area. This term refers to stimulating a peripheral nerve by applying electricity to it, a dorsal root, or a ganglion [3].

The use of electricity in medicine and pain is not new. It dates back to thousands of years BCE when humans learned to explore the use of electrical charges of fishes for the treatment of headache- and gout-related pain [4,5]. In the 1860s, the French scientist G. Gaiffe created the first known transcutaneous electrical nerve stimulation (TENS) device [4]. The use of direct electrical stimulation of a peripheral nerve was first reported around a similar time by Julius Althaus, a German-English physician [6]. In 1919, Charles Kent patented and sold “The Electreat”, which was an electric-massage machine that claimed to improve the health of every organ [7]. Kent was later charged with misbranding. A second wave in the modern development of neuromodulation came in the 1980s when cylindrical leads, implantable pulse generators, and paddle electrodes became available [4]. The third wave in modern development was spearheaded by the development of a percutaneous technique to insert electrodes. The technique was demonstrated by Weiner and Reed to treat occipital neuralgia [8]. Peripheral nerve stimulation has found use in not only treating pain but also in the treatment of complex cases of intractable epilepsy and depression [9,10,11].

## 2. Gate Theory

The gate theory proposed by Melzack and Wall in 1965 suggests that applying non-painful stimuli to the low-threshold, non-nociceptive, large-diameter A beta fibers causes activation of the inhibitory interneurons and inhibition of the nociceptive A delta and C fiber conduction and discharge in the dorsal horn and subsequent transmission to the central cortex [12]. While this theory does explain the foundation of neuromodulation, there remain ample questions about the mechanism of action and the explanation of its analgesic effect [13].

Two of the commonly used techniques, acupuncture and TENS, have been used traditionally to treat pain and share some basic principles. It is common to compare the PNS mechanism with these techniques. Acupuncture utilizes the technique of use of needles to stimulate the nerves at specific points in the body [14]. Cutaneous signals of low current and frequency are conducted to the spinal cord via these needles, where there is an interaction between these peripheral and visceral inputs. This results in not only peripheral neuromodulation but also inhibition of central sensitization. Similarly, the concept of TENS involves the application of various intensities of stimuli to the peripheral nerve via transcutaneous electrodes and causing neuromodulation at both the peripheral and central levels [15,16].

Neuromodulation systems such as spinal cord stimulation (SCS) and PNS deliver electrical pulses to the spinal cord or peripheral nerves, changing the transmission of pain signals to the brain. Figure 1 below depicts the effects of stimulation on nociceptive processing, as well as spinal inhibition, activation of inhibitory system, and eventual cortical stimulation leading to a decrease in pain perception.

## 3. History of PNS

Historically, the first reported use of PNS dates to the mid-1960s [17]. Since then, there has been significant evolution and advancements in the techniques, equipment, and devices used. The concept of PNS involves electrical stimulation of a specific nerve trunk or ganglion through implanted subcutaneous electrodes placed in close vicinity of the target nerve. Peripheral nerves are located beyond the brain and spinal cord. After the electrode is placed near the target peripheral nerve, it is attached to an external generator that allows the delivery of electrical pulses that elicits a paresthesia sensation [18].

The initial attempts at PNS involved surgical dissection and implanting the leads to the nerve [19,20,21]. Wrapping the leads around the nerve and placing a fascial graft between the nerve and lead were some of the techniques used. However, not much popularity was achieved either, and they fell out of use due to the complex procedures. The popularity gained by the percutaneous lead placement in the spinal cord stimulator (SCS) system led physicians to explore its use to target more peripheral nerves [8,22,23]. The use of both cylindrical and paddle leads to treat occipital headaches and other pain conditions has been documented. Although bypassing the surgical complexities with this option, challenges remain related to mechanical stresses on the leads, especially with the joint movements leading to lead fractures and migrations [24,25]. Table 1 lists some of the peripheral nerves where PNS may be implanted.

The recent decade has seen the introduction of newer devices that are minimally invasive and involve percutaneous implantation. The use of ultrasound imaging to guide lead placement has increased the precision in the placement of the leads but also captured more nerves. It has been possible to target large-diameter afferent sensory fibers at large frequencies and efferent fibers at a lower frequency and achieve substantial reduction in pain [26]. Below is a schematic diagram of an implanted peripheral nerve stimulator lead targeting the superior cluneal nerve (Figure 2).

## 4. Mechanism of Action

The concept of peripheral and central sensitization after an injury to the nerve plays a vital role in the development of chronic neuropathic pain. The inflammatory cascade activated due to the injury and release of many pro-inflammatory cytokines and neuropeptides results in the hyperexcitability of nociceptive afferents. This causes not only sensitization of nociceptive-specific (NS) and wide-dynamic-range (WDR) second-order neurons in the dorsal horn of the spinal cord but also a reduction in the inhibitory GABAergic and glycinergic transmission.

Alterations in descending modulation from the periaqueductal gray (PAG) and rostral ventromedial medulla (RVM) lead to heightened transmission of the pain signals to the thalamus and sensory cortex. Expansion of the pain representation in the cortex occurs, leading to aberrant sensory processing [27]. Glial activation, chemical phosphorylation, ectopic firing, and excitation of interneurons act at both spinal and supraspinal levels [28,29,30]. The interplay of all the above factors leads to a decreased threshold, aberrant processing of nociceptive signals, and maintenance of a hyperexcitable state at the peripheral, spinal, and supraspinal levels.

Some investigators believe that the mechanism of PNS responsible for pain relief is in the central nervous system (CNS). In contrast, others suggest that it is a peripheral one involving a conduction block in small-diameter afferent fibers, thereby preventing the arrival of nociceptive information to the CNS.

Still under discussion and a topic of research, alternative theories to gate control theory have attempted to explain the mechanism of peripheral nerve stimulation, including stimulation-induced blockade of cell membrane depolarization, reducing the excitation of C-fiber nociceptors and suppression of dorsal horn activity, reducing hyperexcitability, and long-term potentiation of dorsal horn neurons and depletion of excitatory amino acids, such as glutamate and aspartate, and release of inhibitory neurotransmitters such as GABA [31]. Both animal and human studies continue to be looked into to shed more light on the mechanisms of PNS.

### 4.1. Peripheral Pathway

Chronic pain arising from the peripheral nerve increases the local concentration of mediators such as endorphins and prostaglandins, leading to increased blood flow. The PNS has been shown to down-regulate neurotransmitters, endorphins, local inflammatory mediators, and blood flow at the peripheral level [32]. Electrophysiological studies demonstrate decreased ectopic discharges. As a result, there is a decrease in the transmission of efferent nociception [33].

It has been extensively studied how excitability and conduction velocity of the nerve fibers are subnormal following tetanic stimulus [34]. Additionally, high-frequency stimulation can cause an exponential decline in conduction velocity of both myelinated and non-myelinated nerve fibers [35]. Torebjork and Hallin [36] went on to demonstrate that repeated electrical stimulation of intact radial and saphenous nerves resulted in the excitation failure of A and C fibers. Sciatic nerve stimulation at low to medium frequencies in rats with sciatic nerve injury demonstrated nerve regrowth and change in the local chemical milieu [37].

Swett [38] demonstrated with their study that the analgesic effect of PNS occurs with stimulus intensities above the threshold of perception but below the threshold for pain. Their observation does not support gate theory and rather suggests a central mechanism for the PNS. 

### 4.2. Central Pathway

Consistent with the gate theory, PNS activates A beta fibers in the periphery, which leads to inhibitory dorsal interneurons and inhibition of A delta and C fibers. The literature review has demonstrated that PNS may be involved in modulating higher CNS centers, including the dorsal lateral prefrontal cortex, somatosensory cortex, anterior cingulate cortex, and parahippocampal areas [39,40,41].

The effect on the GABAergic and glycinergic increases in serotonin and dopamine metabolites can result at the spinal level [42,43]. Changes in the levels of the substance P and calcitonin gene-related protein (CGRP) may also play a role [44]. PNS also offers increased inhibition of dorsal WDR neurons [40], decreased A beta fiber activation in the medial lemniscus pathway [40,45,46], and spinothalamic tracts [47].

The trigeminocervical complex (TCC) extends from the trigeminal nucleus caudalis to the segments of C2-C3 and is a convergence from other afferent sources. The nociceptive inflow to second-order neurons in the spinal cord, and the TCC is subject to a modulation by descending inhibitory projects from the brain stem structures such as the periaqueductal gray (PAG), nucleus raphe magnus, and the rostroventral medulla because stimulation of these regions produces profound antinociception [48]. Thalamic activation with PNS has been suggested to occur without a change in the underlying brainstem activation [49].

Animal studies performed in cats support the central pathway where pudendal or tibial nerve PNS increases inhibitory input on bladder-related interneurons [50]. Another rat model shows up-regulation of Arc protein expression and decrease in GluA1 transcription in the dorsal horn, inhibiting neuropathic, inflammatory, and bone cancer pain [51].

## 5. Craniofacial PNS

The ability of the nervous system to adapt to stimuli by reorganizing its structure or functions is known as neuroplasticity. Studies have demonstrated the theory that paired associative stimulation (PAS,) which is a combination of PNS and transcranial magnetic stimulation (TMS), induces long-term changes in the excitability of the cerebral cortex, leading to potential motor recovery in post-stroke patients and patients with amyotrophic lateral sclerosis [52,53]. This theory has been implicated in an *N*-methyl-d-aspartate (NMDA) receptor-mediated plasticity. Blocking the NMDA receptor activity results in analgesia.

Suboccipital PNS likely has a different mechanism of action. In a study by Matharu and colleagues [54], patients with migraine who had positive response to suboccipital stimulation showed significant changes in cerebral blood flow in positron emission tomography (PET) imaging studies, suggesting other mechanisms such as central neuromodulation. Other PET studies have also had similar findings, and they have demonstrated increased cerebral blood flow in the anterior cingulate and insular cortices, anteroventral insula, and thalamus [55].

Functional near-infrared spectroscopy (NIRS) and functional MRI (fMRI) are also being utilized to study the significant effects of PNS [41]. Electroencephalography (EEG) and fMRI studies have reinforced that dorsal column stimulation induces changes in cortical activation throughout many regions making up the pain matrix and is hypothesized to activate the descending pain inhibitory system through the modulation of the pregenual anterior cingulate cortex [56,57]. Given the significant role that cortical processes play in producing and potentially reducing chronic pain, the new theory of peripherally induced reconditioning of the central nervous system may help to explain sustained relief following PNS [26].

The mechanism of PNS is likely a combination of both peripheral and central pathways [58]. While large-diameter sensory fibers may directly engage the gate mechanism to decrease pain signals, large motor fiber activation may be translated into physiological neural afferent signals that help gate or decrease pain. Decreasing pain signals over an extended period allows PNS therapy to possibly disrupt the cycle of centrally mediated pain, resulting in activity-dependent neuroplasticity to maintain decreased pain long after the active stimulation period [59].

For some patients, blocking nociceptive afferent input is sufficient to transiently alter the cortical reorganization and reduce chronic pain. However, nerve block has no effect or only a transient effect for others, indicating that although some cases of cortical sensitization rely on continued peripheral input, others appear to be centrally maintained [60]. Robust non-nociceptive afferent input to the cortical areas representing the painful focal region may reduce the severity of pain by actively reconditioning the CNS from the periphery, as opposed to the passive deprivation of nociceptive input that may occur as the result of nerve blocks or ablation. This process has been termed “reconditioning” [61,62].

Occipital nerve stimulation (ONS) has provided insight into the mechanism of PNS for craniofacial pain. The occipital nerves interconnect with the ophthalmic division of the trigeminal nerve and form a neural network that affects the trigeminal nucleus caudalis and cervical dorsal horn at the C1 and C2 levels [63]. This is known as the trigeminocervical complex (TCC). The nociceptive inflow to second-order neurons in the spinal cord and the TCC is subject to modulation by descending inhibitory projects from the brain stem structures such as the periaqueductal gray (PAG), nucleus raphe magnus, and the rostroventral medulla because stimulation of these regions produces profound antinociception [48]. Thalamic activation with PNS occurs without a change in the underlying brainstem activation, suggesting a neuromodulatory mechanism for PNS therapy [49]. Another theory is that occipital nerve stimulation may modulate nociceptive stimulation by increasing extracellular gamma-aminobutyric acid and thus decreasing glutamate levels [64]. The success of PNS in reducing headache syndrome can be attributed to the reduction in afferent inputs into the trigeminal nucleus caudalis and high cervical dorsal horns [48].

A study by Kovacs et al. using fMRI to evaluate CNS activity in occipital nerve stimulation supports that ONS has activating and deactivating effects centrally. The predominant affected areas seen on fMRI were the hypothalamus, thalami, orbitofrontal cortex, prefrontal cortex, periaqueductal gray, inferior parietal lobe, and cerebellum. The areas that were deactivated were primary (M1, V1, A1 and S1), the amygdala, paracentral lobule, hippocampus, S2, and SMA. PET scan studies by Matharu et al. [54] showed that suboccipital stimulation was associated with changes in regional cerebral blood flow in the dorsal rostral pons, anterior cingulate cortex, cuneus, frontal cortex, thalamus, basal ganglia, and cerebellum. These areas in the brain are associated with pain. 

In patients with fibromyalgia, stimulation at the level of the second cervical vertebra or C2 has been thought to exert effects on the autonomic nervous system. Neurons at the C2–C3 level may also exert influence on lateral spinothalamic pathways [65]. ONS may cause changes in the anterior cingulate per PET and EEG studies and, therefore, may affect dopaminergic modulation of the area [66]. Dopamine has been implicated in the pathophysiology of fibromyalgia [67].

Cervical vagus nerve stimulation (VNS) has shown benefits in treating migraines and chronic inflammatory diseases such as rheumatoid arthritis and Crohn’s disease. It also has been studied in epilepsy. Vagus nerve stimulation is believed to also modulate peripheral and central nociceptive functions. It inhibits inflammatory cascades in the periphery through reduced TNF alpha, IL-1Beta, IL-18, HMGB1 protein, and other cytokines [68]. The vagus nerve carries several different fibers, including general somatic afferent, special somatic afferent, general visceral afferent, and general visceral efferent fibers [69]. The vagus nerve transmits afferent signals from peripheral inflammation to the brainstem at the nucleus of the solitary tract (NTS), where 95% of vagal afferents project. The NTS projects to the medial reticular formation, hypothalamus, thalamus, amygdala, hippocampus, and dorsal raphe. Imaging studies suggest decreased thalamic activity centrally due to VNS [69].

Trigeminal nerve stimulation has been demonstrated to show benefits for trigeminal neuropathic pain. Studies involving PET and trigeminal ganglion stimulators for trigeminal neuralgia showed increased CBF in the ipsilateral superior parietal and superior frontal cortices. They decreased CBF in the cerebellum and medial orbitofrontal cortex in the short and long term [70]. Compared to short-term stimulation, long-term trigeminal nerve stimulation showed increased CBF in the orbitofrontal and medial frontal cortices that extend into the rostral cingulate cortex and decreased metabolism in the caudal ACC. This suggests a central role for medial frontal and anterior cingulate cortex involvement in pain modulation of trigeminal neuralgia [69]. 

The supraorbital nerve has been targeted as an area for stimulation to treat cluster headaches and migraines. The supraorbital nerve is a branch of the V1 branch of the trigeminal nerve. Current knowledge of migraines suggests there is neuronal hyperexcitability resulting in trigeminovascular activation. Supraorbital nerve stimulation is thought to decrease the excitability of trigeminal pain pathways through altered activity within the trigeminovascular system and peripheral and central nervous systems [70].

The therapeutic field of SCS involves multiple waveforms such as burst, tonic, high-frequency, and differential target multiplexed. PNS technology typically delivers low-frequency and tonic stimulation to stimulate sensory fibers [71]. Different waveforms allow different stimulation depending on the patient’s pain but still allow for adjustment for the patient’s comfort and tolerability. Unique waveform modalities continue to be developed and optimized to meet patients’ needs. PNS stimulation methods are similar to conventional SCS stimulation, i.e., inducing a “comfortable paresthesia” in the region of pain with 50–100 Hz [72,73].

## 6. PNS in the Acute Perioperative Setting

Despite the widespread use of PNS in patients with chronic pain, the popularity of PNS in the acute postoperative setting is limited. Commonly, postoperative pain management involves using non-steroidal anti-inflammatory drugs, acetaminophen, and opioids, as well as interventional procedures such as peripheral nerve blocks (single or continuous via catheter), among others. While conventional methods are very effective in controlling postoperative pain, it is not without risks. Postoperative opioids predispose patients to adverse effects such as nausea, vomiting, respiratory depression, and sedation. Peripheral nerve catheters, while offering a longer duration of pain relief (i.e., 3 to 7 days), have some limitations, which include infection, catheter dislodgement, local anesthetic toxicity, and dense motor/sensory blockade [74]. Having to carry around a bag filled with local anesthetic upon hospital discharge can also be cumbersome for some patients. Recently, the US Food and Drug Administration (FDA) approved using the Sprint PNS system for use in the setting of acute pain.

One of the earliest studies employing PNS in the acute postoperative setting was performed in patients who underwent total knee arthroplasty [75]. As mentioned earlier, the mechanism of action is largely explained by the gate control theory wherein large-diameter myelinated afferent peripheral nerve fibers are triggered with the use of electrical current and impede pain signals to the central nervous system from small-diameter pain fibers at the level of the spinal cord [76,77]. PNS leads were placed either in the femoral nerve (for anterior knee pain) or sciatic nerve (for posterior knee pain) using ultrasound guidance. Stimulating the femoral and sciatic nerve trunks rather than the distal branches allows less leads to cover a much larger area and enables the leads to be farther away from the surgical field, thus decreasing the chances that it will interfere with the surgery and theoretically reducing the risk of infection. Outcomes were assessed within approximately 2 h since placement and were found to decrease pain by an average of about 63%. PNS enables a passive range of motion of the knee with less pain [76]. Another preliminary study following PNS placed percutaneously prior to anterior cruciate ligament reconstruction surgery has also suggested efficacy in providing analgesia and decreasing opioid requirements [78].

Other potential uses of PNS in the acute postoperative stage include ambulatory foot surgery, specifically bunionectomy surgery. In a study by Ilfeld and colleagues in 2018 [77], a PNS lead was placed posterior to the sciatic nerve under ultrasound guidance preoperatively. This lead was left in place until postoperative days 14 to 28. During this period, the pain scores averaged less than 1 on the numeric rating scale. However, lead fracture and dislodgement were major concerns.

The use of PNS in upper extremity surgery, such as rotator cuff repairs, has not demonstrated appreciable postoperative analgesia when placed to stimulate the suprascapular nerve. However, it provided satisfactory results during the first two weeks following surgery [78].

Currently, preliminary studies have suggested the utility of PNS in the acute postoperative setting; however, further large-scale, prospective, randomized trials are needed to explore its efficacy and appropriate nerve targets in acute pain management. 

## 7. Current Available Devices

The currently available percutaneous PNS systems available in the market are Stimrouter (Bioness), StimQ (Stimwave), Sprint, Reactiv8, and Nalu.

The StimRouter PNS system (Bioness, Inc., Valencia, CA, USA) is US FDA-approved for the treatment of peripheral mononeuropathy that is refractory to conservative management [79]. This system consists of a flexible tined lead with a stimulation and receiver end, an external pulse transmitter, disposable electrodes, and a patient programmer (to turn stimulation on/off or select stimulation intensity and program). For chronic mononeuropathy-related pain, it can be placed in numerous locations such as the arms, trunks, and legs. Among the implanted nerves are axillary, genitofemoral, intercostal, ilioinguinal, lateral femoral cutaneous, peroneal, saphenous, suprascapular, sural, and tibial [80]. A case series consisting of 39 patients showed significant reduction in pain scores and opioid consumption, with the most observed success in the brachial plexus (80%) and suprascapular nerve (80%), and the smallest success in the intercostal nerve (40%) [81]. Once implanted, the lead is MRI conditional, while the rest of the external components are removed prior to entering the MRI suite.

The Sprint PNS system (SPR Therapeutics, LLC, Cleveland, OH, USA) has been used for acute or chronic axial low back pain [82], shoulder pain [83], and post-amputation pain [84,85]. This system is FDA-cleared for percutaneous stimulation for up to 60 days before the removal of the leads.

The advantage of this system is that it aims to reduce infection rates and lead migration by using a flexible open-coil lead placed remotely from the target nerve [86,87]. In a retrospective review of 4481 patients implanted with the Sprint PNS system from 2017 to 2021, 72% of patients had 50% or greater pain relief and quality of life improvement across nerve targets [88]. Several other studies have shown success with this system as well [82,89]. This system is not MRI-compatible.

The StimQ (Stimwave Technologies, Inc., StimQ PNS System, Pompano Beach, FL, USA) is another PNS system used for chronic pain mediated by a peripheral nerve. This is, however, not intended for pain in the craniofacial region. This system consists of a receiver kit, the StimQ Wearable Antenna Gear (SWAG) Transmitter kit, and an accessory kit. The accessory kit is designed to be worn on the extremities (arms or legs) and on the torso. This system is MRI-compatible.

The Nalu Neurostimulation System (Nalu Medical, Carlsbad, CA, USA) features a 27-times smaller implantable pulse generator (IPG) in the market, along with a battery-free, replaceable, wearable IPG that eliminates implanted battery and battery replacement surgeries in the future. This system is FDA-cleared with an expected service life of 18 years. As with other PNS systems, its use is indicated in adults with severe pain of peripheral nerve origin, except pain in the craniofacial region. Pain after back, hip, hernia, and knee surgeries, as well as nerve pain, can be managed by this system [90,91]. This system is MRI-compatible up to 3.0 Tesla.

An implantable system aiming to treat multifidus muscle dysfunction is the ReActiv8 Implantable Neurostimulation System (Mainstay Medical, Ltd., San Diego, CA, USA). A rehabilitative as well as a restorative system, ReActiv8 is FDA-approved for bilateral stimulation of the L2 medial branch of the dorsal ramus as it crosses the transverse process at L3, to target multifidus muscle dysfunction by eliciting episodic, isolated contractions, eventually facilitating recovery from low back pain. This system consists of an IPG with two leads, four electrodes, and a patient activator (handheld control unit that communicates with the IPG). It is intended for at-home therapy for 30 min twice daily while the patient is in a prone or lateral position. Eighty-three percent of participants who have this system reported significant improvements in pain, disability, or both three years post-implantation [92]. This system is not MRI-compatible once implanted.

## 8. PNS Safety

Due to the popularity of neuromodulation, the FDA held a symposium on safety and clinical efficacy of implanted neuroaugmentive devices in 1977 [93]. Neuromodulation was deemed safe and effective at the symposium. Safety considerations regarding peripheral nerve stimulation involve assessment of biostability (passive electrode preservation and maintained functionality) and harmlessness (healthy body tissue survival) [94]. Extraneural electrodes have shown superior long-term stability compared to intraneural electrodes [94]. Safety limitations regarding pulse amplitude may vary depending on several factors including electrode placement, size and proximity to the stimulated fiber. Chronic stimulation frequencies below 30 Hz and effective stimulation time below 50% are considered safe, although data are limited [94]. Chronic continuous peripheral nerve stimulation at frequencies above 30 Hz needs further study. The mechanism by which PNS causes neural damage is still unclear. The exact PNS settings that result in stimulation-induced depression of neuronal excitability and neuronal damage also need further study. Clinically, common device-related complications include migration of the stimulator lead, lead fracture, and device malfunction [95,96]. Other complications that are common with similar invasive procedures include infection, bleeding, surgical site pain, and nerve injury. 

## 9. Future Directions

With the current opioid epidemic, it is important to recognize potential nonpharmacological therapies to treat chronic pain. PNS technology continues to be an effective modality, and further investigations on its use in trauma, stroke, myocardial ischemia, and anxiety are warranted. Different waveform modalities and stimulation parameters continue to be studied to optimize patient’s satisfaction. Large, prospective randomized trials to further elucidate its short- and long-term efficacy are still lacking. 

## 10. Conclusions

The mechanism of action of PNS still remains a topic of discussion. Based on animal and human studies currently available, it would be agreeable that both peripheral (altering nerve fiber conduction and excitability) and central (altering neurotransmitters, modulating expression of neuronal signaling proteins, altering activity in central pain matrix regions, or descending inhibitory pathways) factors play a role in the analgesic effect of PNS. Further human studies are needed to further elucidate its exact mechanisms. This is essential not only to further our existing knowledge but also to help in engineering newer devices and technology to improve outcomes.

## Figures and Tables

**Figure 1 ijms-24-04540-f001:**
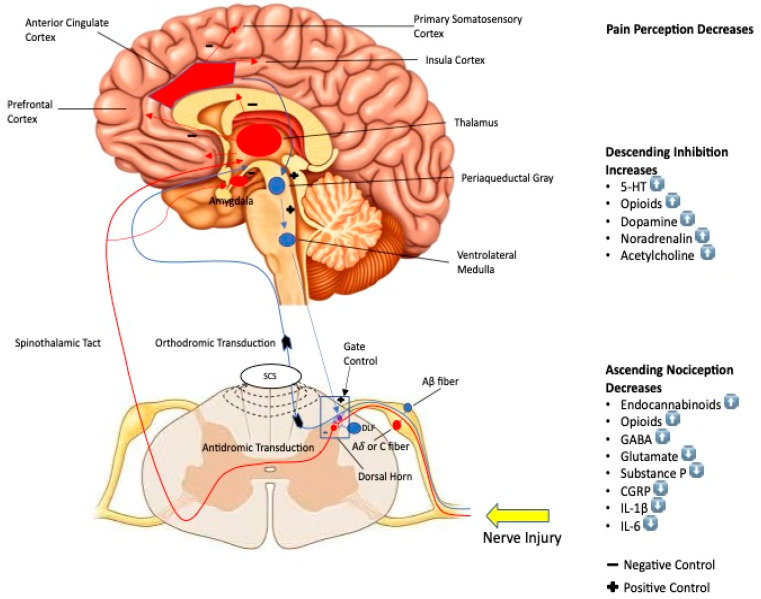
Schematic diagram depicting effects of stimulation on the brain. UP arrow means increase. DOWN arrow means decrease.

**Figure 2 ijms-24-04540-f002:**
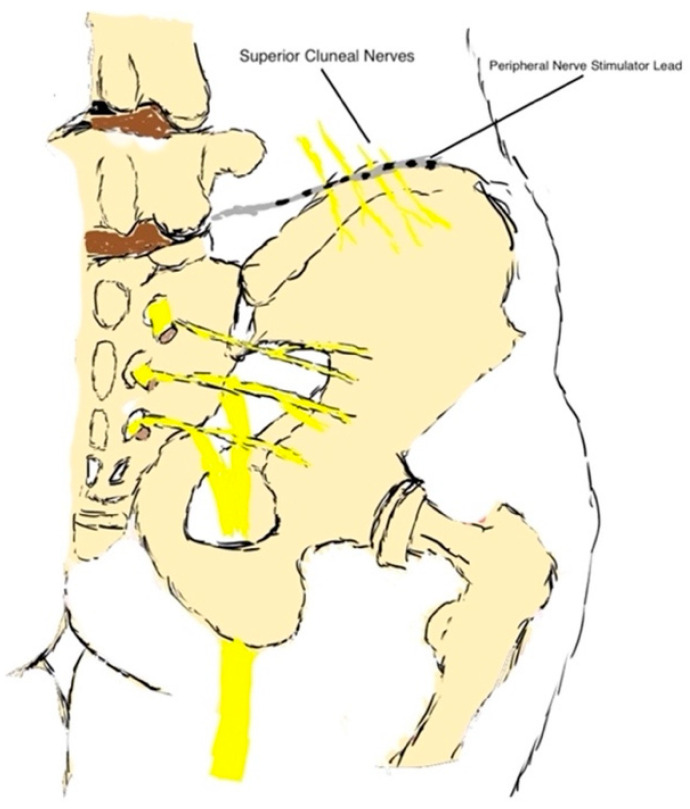
Implanted peripheral nerve stimulator lead.

**Table 1 ijms-24-04540-t001:** Peripheral Nerves Commonly Used for PNS.

Head/Neck	Greater Occipital Nerve
Upper Extremities	Brachial plexus Suprascapular nerve Axillary nerve Radial nerve Median nerve Ulnar nerve
Lower Extremities	Sciatic nerve Obturator nerve Femoral nerve Lateral femoral cutaneous nerve Genicular nerve Saphenous nerve Common peroneal nerve Tibial nerve Sural nerve Superficial peroneal nerve
Abdomen/Trunk/Back/Pelvis	Medial branch nerve Ilioinguinal nerve Iliohypogastric nerve Genitofemoral nerve Cluneal nerve Pudendal nerve

## Data Availability

Not applicable.

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
