# Peer review of "Mechanism of Action of Peripheral Nerve Stimulation for Chronic Pain: A Narrative Review"

_ijms, 2023, doi:10.3390/ijms24054540_

Round 1

Reviewer 1 Report

General comments:

Although this narrative rather comprehensive review of “mechanisms of action of peripheral nerve stimulation” covers a wide range of possible applications and to some extent gives a short description of the mechanism of action it is sometimes hard to follow as results from previous studies are basically just listed one after the other without giving any deeper insight into the pros and cons for the particular application. 

While table 1 is informative the manuscript has an apparent lack of figures describing the actual  “Mechanisms of Action” for any of the pathways of molecular responses during and after stimulation.

An addition of such figures/diagrams together with additional subheadings (especially from page 5 onwards) would greatly improve the quality of the manuscript.

The conclusion could be expanded to at least cover some aspects of “action of mechanisms”.

Specific comments:

Abbreviations should be used in a consistent manner which is not always the case.

I find some statements lacking references such as on page 4 first paragraph and line 220 page 6 among others.

The only figure provided merely show an “Implanted peripheral nerve stimulator led” and is of little value if the topic is (again) “Mechanisms of action”. In my copy the figure is also of very poor quality.

My recommendation would be to restructure the text, insert subheadings where justified and expand on the actual mechanisms. For example, statements like that on line 164/page 5 provide no information of how PNS results in enhanced NMDA-mediated plasticity. 

My final recommendation would be reconsidering after major revisions.

Author Response

Please see attachment. Track changes used to show new edits/changes.

Reviewer 2 Report

I find this review very well done and suggest it be published as-is.   The authors write very well and present the present state of the art in a comprehensive manner and do a good job on the citation of references.

I have no real criticisms of this paper.

Author Response

Thank you for your comments.

Reviewer 3 Report

The authors provide a comprehensive review on the mechanisms and hypothesis behind neuromodulation for the prupose of pain mangement. Indeed, there is little high quality evidence on this topic. Nevertheless, the paper is well written, various aspects of neuromodulation and possible mechanisms are discussed in this review. However, it should be clarified in the title, that the review is limited to pain management. 

Author Response

Thank you for your comments.

Title is changed to the following:

Mechanism of Action of Peripheral Nerve Stimulation for Chronic Pain:

A Narrative Review

Round 2

Reviewer 1 Report

The manuscript is substantially improved after revision. I recommend acceptance in present form.